# Can LLMs *really* reason about Code?

## Studying how well LLMs understand the relation between Input, Code, and Output

## Abstract

In the past years, large language models (LLMs) have demonstrated remarkable progress in code generation. However, their ability to reason about program behavior remains an open challenge—an ability that is relevant for applications including reverse engineering, debugging, secure code generation, test-driven synthesis, input reconstruction, reverse fuzzing, behavioral monitoring, and safe execution modeling.

To study this ability, we examine the capacity of LLMs to reason about the *semantics of code*—specifically, their ability to *relate* code, its inputs, and its outputs to each other. To this end, we investigate whether and how well LLMs can predict one of these three components given the other two—that is, (1) predict the *input* given code and output, (2) predict the *output* given code and input, and (3) predict the *code* given input and output. This way, we assess how well LLMs can reason about and understand the underlying relationships that govern program execution.

We construct four datasets covering string processing, array operations, and coding challenges in JavaScript and Python to evaluate diverse program-understanding capabilities, incorporating various code mutation techniques to increase complexity.

In our evaluation on tasks covering string processing, array operations, and coding challenges, we find that *closed-weight models* achieve the strongest performance across all datasets, including perfect input recovery on deterministic string tasks. Across tasks, *output prediction* is comparatively stable, whereas *code prediction* remains the hardest setting and often fails for smaller models. Finally, *cross-codebase* transfer is feasible, especially for input prediction, but highly sensitive to model capacity and fine-tuning strategy.

## CCS Concepts

• **Computing methodologies → Learning from demonstrations**; **Neural networks**; • **Software and its engineering →** *Software testing and debugging*.

## Keywords

code generation, behavior modeling, reverse engineering

**ACM Reference Format:**
Anonymous Author(s). 2026. Can LLMs *really* reason about Code?: Studying how well LLMs understand the relation between Input, Code, and Output. In *Proceedings of Make sure to enter the correct conference title from your rights confirmation email (Conference acronym ’XX).* ACM, New York, NY, USA, 9 pages. https://doi.org/XXXXXXX.XXXXXXX

## 1 Introduction

LLMs have advanced remarkably over recent years in solving code-generation problems, with these models capable of handling a wide

*Conference acronym ’XX, Woodstock, NY*
2026. ACM ISBN 978-1-4503-XXXX-X/2018/06
https://doi.org/XXXXXXX.XXXXXXX

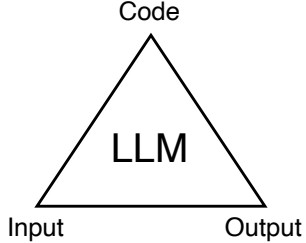

Code

LLM

Input          Output

**Figure 1: The LLM Triad: To examine how well LLMs understand program semantics, we give them two triad components and let them predict the third component.**

range of programming tasks, from solving simple programming challenges to mocking a (simple) C compiler [19, 30]. Despite their impressive generative abilities, it remains unclear to what extent LLMs understand and reason about program behavior [4], a fundamental aspect of computer science that encompasses the ability to determine how a program transforms inputs into outputs and to reason about correctness, edge cases, and failures—capabilities that go beyond mere code generation and venture into program comprehension. To explore this ability, we introduce a set of experiments designed to evaluate the ability of LLMs to reason about the *semantics* of a program—specifically, the relationship between the code, its inputs, and its outputs (Figure 1).

For example, consider a string manipulation function `CamelCase()`, which transforms an input string `"bird flight"` into its camel case form `"birdFlight"`. In this scenario, we investigate

(1) whether the model can infer a correct *input,* `"bird flight"`, when provided with the `CamelCase()` code and the output `"birdFlight"`;
(2) whether LLMs can accurately predict the *output* given both the code and the input, without executing `CamelCase()`; and
(3) whether the model can generate a suitable function body for `CamelCase()` when given the `"bird flight"` input and `"birdFlight"` output.

While this is a relatively simple example, it serves to illustrate the broader challenge. By systematically mutating the code (and thus synthesizing new code examples not seen during training), we push the model beyond pattern matching and prompt it to truly reason about the structure and semantics of code.

The ability to predict code, program inputs, or outputs could have significant implications for several real-world applications:

**Predicting Code.** In *reverse engineering* scenarios, we could derive the underlying source code from input-output behavior to reconstruct legacy or undocumented systems. We could *produce secure and maintainable code* from high-quality datasets, improving software robustness and reducing vulnerabilities. And we could *generate code that satisfies given*

*input-output test cases*, enabling AI-assisted problem-solving in education, automation, and development workflows.

**Predicting Outputs.** By predicting expected outputs based on code and inputs, we could *detect anomalies* during execution, identifying potential bugs or security issues. As models would predict behavior without execution, we could *model safe execution* of untrusted or potentially malicious code.

**Predicting Inputs.** Given observed code and outputs, we could deduce likely input values to assist with *debugging*, testing, and behavioral analysis. Likewise, we could *generate test cases* to produce specific outputs or traverse certain execution paths, improving bug detection and test efficiency.

To assess these prediction abilities, we construct four datasets covering string processing, array operations, and coding challenges in JavaScript and Python to evaluate diverse program-understanding capabilities, incorporating various code mutation techniques to increase complexity. We formulate program comprehension as a triadic completion problem: given any two of {code, input, output}, predict the third.

We evaluated this problem set on *open-weight models*, such as Phi-4, Qwen3, and *closed-weight models* from the OpenAI GPT family on three mutation-augmented JavaScript datasets (Voca: 17,165; Stdlib Strings: 3,595; Stdlib Arrays: 4,429) and one mutation-augmented Python dataset derived from LiveCodeBench (6,820). For open-weight models, we performed task-specific fine-tuning; all models are evaluated on held-out samples using a combination of text similarity (BLEU) and execution-based functional checks tailored to each subtask.

Our results indicate a consistent *performance gap* between closed- and open-weight models. Closed-weight models achieve perfect *input prediction* on deterministic string transformations (e.g., 100% on Stdlib Strings) and strong *output prediction* (roughly 70–80% on Voca and Stdlib), whereas code prediction remains the most challenging subtask: most models stay below 30% accuracy, with the best results on the Python LiveCodeBench-derived dataset. Our results suggest that closed-weight models have a decent grasp of program semantics, while open-weight models still struggle to fully capture the underlying relationships; abstracting concrete examples into general code remains a key challenge for LLMs.

## 2 Background

*LLMs for Code Generation.* Recent advancements in code generation enabled LLMs to produce code that is syntactically and semantically valid. Models such as CodeT5 [27], InCoder [5], and PolyCoder [28] are trained entirely on code from the ground up and are designed to perform program-related tasks, including code completion, translation, and summarisation. In contrast, fine-tuned code models such as Codex [10] and StarCoder [13] are derived from general-purpose LLMs (e.g., GPT) and adapted to code datasets, enabling them to bridge natural language prompts with formal code generation more effectively. While code LLMs have achieved considerable success, they are typically designed for unidirectional tasks, where natural language is the sole input for generating code.

*Reasoning from Input-Output Examples.* In contrast, other approaches focus on learning from input-output pairs. For example, RobustFill [7], based on a modified attention RNN architecture,

achieves up to 92% accuracy in generating programs from input-output examples. Similarly, DeepCoder [3] demonstrated early success in synthesizing small programs by leveraging neural networks to reason over input-output pairs within domain-specific languages. CodeI/O [12] introduces a method for learning general reasoning patterns by converting Python functions into input-output pairs accompanied by natural language explanations. Chain-of-thought rationales describe the underlying logic, allowing LLMs to internalize structured reasoning from code. An enhanced version, CodeI/O++ [12], further refines this process through iterative verification and correction. While their approach achieves strong results across symbolic, logical, and commonsense reasoning tasks, it relies heavily on natural language prompts, unlike our method, which learns directly from input-output behavior without depending on verbal explanations.

*Code reasoning.* Predicting a program's output from given code and inputs is perhaps the most direct form of program reasoning. Prior work has tested LLMs on benchmarks like HumanEval [10] and MBPP [2], where models are asked to generate code that matches target outputs for specific inputs. Though not always framed as output prediction, passing test cases indirectly implies the ability to model input-output behavior.

*Behavior Modeling.* Our study is inspired by *Modelizer* [15], a framework for learning reversible models from the input-output behavior of black-box systems using sequence-to-sequence translation techniques. While Modelizer does not consider the program code beyond its execution, our work extends these ideas by extending the learning context to a triadic prediction setting. Instead of training a neural network from scratch, we use pre-trained open- and closed-weight models. To the best of our knowledge, the triadic prediction setting has not been explored to date.

Several other systems have also tried to model system behavior using LLMs. Ding et al. have proposed *TRACED* [9], an approach to train execution-aware models using combinations of source code, executable inputs, and code execution traces. The *SEMCODER* [8] Code LLM relies on a *monologue reasoning* principle, inspired by rubber-duck debugging when the model "talks to itself" in natural language to explain and understand the code comprehensively, mimicking how a developer verbally reasons through code. *CodeExecutor* [14] is another Transformer-based architecture designed to model and predict program execution traces using solutions drawn from programming challenge datasets.

## 3 Methodology

To evaluate the ability of LLMs to solve the LLM-Triad, we need suitable datasets. Such datasets must include the required components—input, output, and code—while minimizing overlap with models' pretraining corpora to reduce the risk of benchmark contamination. Most existing datasets focus on *code generation* rather than code execution or program-level reasoning. Furthermore, existing benchmarks predominantly focus on Python, a language already well represented in LLM pretraining data. As a result, models may rely on memorized patterns and language-specific familiarity

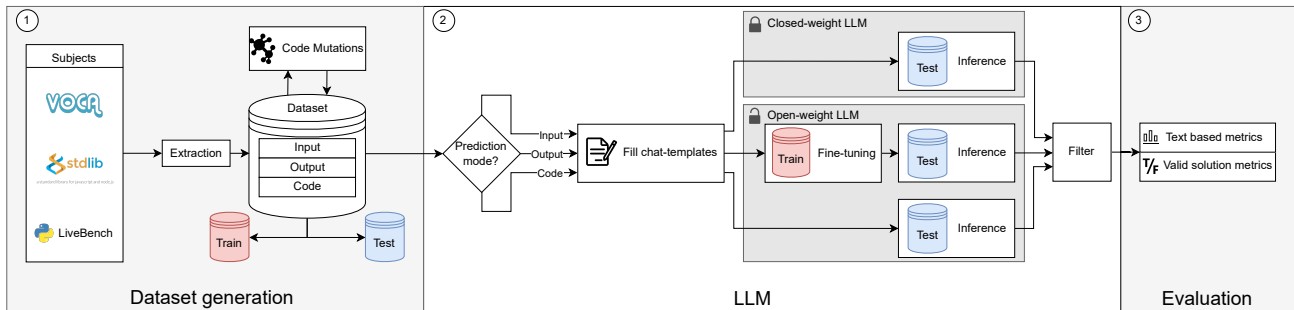

**Figure 2: Experiment pipeline.** ① **Mutated datasets from Voca, Stdlib, and LiveBench are split into training and test sets.** ② **Closed- and open-weight LLMs predict code, input, or output; open-weight models are fine-tuned before inference, and predictions are filtered.** ③ **Results are evaluated using text-based and solution-validity metrics.**

instead of demonstrating genuine reasoning about program behavior. To address these limitations, we construct several JavaScript-focused datasets, which we assume are comparatively underrepresented in existing benchmarks. In addition, we derive a Python dataset based on LiveCodeBench [11] to introduce novel challenges that are unlikely to be present in LLM pretraining corpora. Our datasets are built around functions from selected libraries, with the primary dataset, Voca [23], covering a substantial portion of the library. Each function is self-contained and depends only on its internal logic and explicit imports, eliminating the need for external libraries or runtime modifications.

To further enhance the dataset's novelty and diversity, we apply *code-mutation* techniques to expand it and reduce the likelihood that models encounter memorized code fragments from their pretraining corpora. After dataset construction, the samples are partitioned into training and test sets (see Figure 2 ①).

We then evaluate a selection of LLMs, including both open- and closed-weight models, on their ability to address the subtasks of the LLM-Triad. This comparison allows us to assess whether open-weight models can achieve performance comparable to proprietary alternatives and whether both model types can effectively solve the individual subtasks. *Open-weight models* are particularly attractive because they require only computational resources and can be fine-tuned at relatively low cost. This flexibility enables domain-specific adaptation, which is often unavailable or prohibitively expensive for closed-weight systems. Therefore, fine-tuning is applied exclusively to open-weight models to leverage their adaptability, while ensuring that all fine-tuned models are evaluated exclusively on unseen data (see Figure 2 ②).

Each model is evaluated in a single-task setting, where it predicts one component at a time—input, output, or code—given the remaining components. The predictions are subsequently evaluated using both text-based similarity metrics and functional correctness measures to assess solution validity (see Figure 2 ③).

## 3.1 Datasets

*3.1.1 Voca JavaScript.* We use Voca [23] as the primary JavaScript codebase. Voca is a widely used and long-maintained string manipulation library (60+ functions such as *trim*, *split*, and *camelCase*). Its long development history makes it likely to be human-written,

reducing the risks associated with LLM-generated training data [24]. Voca's functions are largely self-contained and do not rely on external dependencies, providing a clean setting for controlled experiments [26].

In this study, the target code corresponds to a single logical unit. Imports and constants are excluded from the target code and instead provided separately as structured prompt context, supplemented by brief mined comments, see Figure 3. We generate single-edit code mutations using Stryker [25] (e.g., *EqualityOperator*, *ConditionalExpression*, *LogicalOperator*, *ArithmeticOperator*, *AssignmentExpression*), creating one mutated variant per mutation site. Inputs are taken from Voca's existing test suite, and outputs are computed by executing each mutated function with Node [21]. Splits are performed either by holding out entire base functions (testing generalisation to unseen functions) or by holding out input–output pairs (testing behavioral generalisation within seen functions), targeting an approximate 80–20 split. The dataset contains 17,165 entries.

*3.1.2 Stdlib Strings JavaScript.* The second dataset is constructed around the JavaScript Stdlib library. This library provides functionality equivalent to Voca, but with a different internal implementation. We select ten functions from this library that are also present in Voca, namely *camelcase*, *endsWith*, *kebabCase*, *lowercase*, *lpad*, *repeat*, *replace*, *reverse*, *slice*, and *snakecase*. This overlap enables cross-evaluation, allowing us to assess how well the models generalize to identical behavior implemented in different ways. We follow the same dataset structure as for Voca and extract the selected functions in a function-only code format. The corresponding imports, along with comments describing their functionality, are included in the prompt. In contrast to Voca, the JavaScript standard library exhibits a substantially different structure and implementation. Code permutations are generated using Stryker, and the function inputs are identical to those used for Voca. The training and test split is again performed at the level of seen and unseen functions, and the same selection is done as in Voca. This is necessary; otherwise, when cross-evaluating, the fine-tuned LLM would already have seen the input-output pairs for this specific function. The dataset contains 3,595 entries; Figure 3 shows an example.

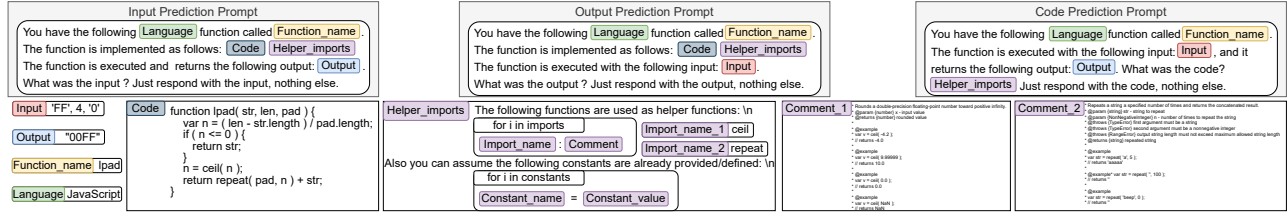

**Figure 3: Example of a Stdlib String dataset entry (*lpad* function) and the prompt design used in our input, output, and code prediction experiments. Placeholders are replaced with dataset-specific entries during prompt construction. The entry includes its input, output, code sample, and imports. No additional constants are associated.**

*3.1.3 Stdlib Arrays JavaScript.* The third dataset is also based on the Stdlib library; however, instead of string manipulation functions, it focuses on array construction and modification utilities namely: *fullLike*, *incrspace*, *mskfilter*, *nans*, *oneToLike*, *place*, *slice*, *take*, *typedarray2json*, and *zeroTo*. This choice increases the overall task complexity, as array-based operations involve substantially more arithmetic computation and index-based reasoning. It is well established that LLMs handle string-based transformations comparatively well. The inputs for the array modification functions are generated using an LLM. The model is prompted to produce additional inputs based on the function description, input parameters, and a small set of examples. To limit the dataset size and reduce computational cost and runtime, we generate 10 sample inputs per function before mutation and filter out incorrectly produced inputs. The remainder of the dataset construction process follows the same methodology as for Voca and the standard library datasets, including data splitting, mutation generation, and other steps. The dataset contains 4,429 entries.

*3.1.4 LiveCodeBench Python.* The final dataset is derived from the LiveCodeBench benchmark [11], which was originally designed to evaluate code generation capabilities. LiveCodeBench consists of Python functions, each with one example input and output, accompanied by a natural-language description, and is continuously updated with newly generated samples that are excluded from model pretraining to ensure evaluation on previously unseen tasks. From this benchmark, we extract a subset of 80 distinct functions. For each function, we generate ten mutated variants using the Mutmut tool[1]. In addition, following the methodology applied to the Stdlib dataset, we generate additional input samples using GPT, as LiveCodeBench provides only a single test case per function. To control the overall dataset size and maintain balance across functions, we limit the number of generated input samples to ten per function. The dataset contains 6,820 entries.

## 3.2 Adaptation of LLMs

We evaluate both open- and closed-weight models under comparable conditions. Each model is first assessed in its standard, pretrained form to establish a baseline of predictive capabilities. Beyond this, we fine-tune the selected open-weight models to better adapt to our target task and dataset. Fine-tuning is particularly valuable for exploring whether models can internalize domain-specific patterns, thus improving performance on specialised subtasks of the

LLM-Triad. We primarily train models on a specific prediction task (e.g., input/output, code/output, or input/code), where the model must infer the missing component from the other two.

*3.2.1 Open-Weight Models.* Given that modern LLMs range in size from a few billion to several hundred billion parameters (e.g., 500B), efficient training and deployment are only feasible on large-scale computing clusters. For the inference and fine-tuning of open-weight models, we primarily employ the Unsloth framework [6]. which is a high-speed training toolkit designed for an efficient fine-tuning of open-weight LLMs in low-resource environments. We restrict the set of selected models to those that can be executed on a single GPU after quantizing the model weights. For our experiments, we considered the following open-weight models: *Phi-4* [1] (14B), and *Qwen3* [29] (4B and 32B).

For all models, we used the most recent model checkpoints prepared by Unsloth as of the end of 2025. Given the fast pace of LLM development, newer versions may exist by publication time. Together, these open-weight models reflect complementary philosophies: scaling for data-centric design (Phi-4), generality and bilingual optimisation (Qwen3). This allows us to probe whether different design choices influence performance on specialized evaluation tasks. In this study, we deliberately avoid extensive hyperparameter exploration. Instead, we rely on the default fine-tuning and inference configurations [6] prepared by Unsloth for all models, ensuring consistency and comparability across experiments.

*3.2.2 Closed-Weight Models.* Closed-weight models are typically inaccessible or quite expensive to fine-tune, but they serve as state-of-the-art references against which open-weight models can be compared. Our dual focus on open- and closed-weight systems is motivated by questions of whether open-weight models can approximate or match proprietary alternatives in the given study scope. Whether open-weight models can be effectively fine-tuned to address the subtasks of the LLM-triad, and whether finetuned open-weight models can outperform the closed-weight models. We evaluate a set of closed-weight models from OpenAI, such as *ChatGPT 5.1* [20], *ChatGPT 4.1* [17], and *o4-mini* [18]. They served as strong baselines for our comparisons. The presence of a smaller closed-weight model (o4-mini) aligns with our broader investigation of scale, efficiency, and sufficiency, as it provides a natural comparison point for open small-scale models such as Phi-4. To simplify the evaluation process, during prompting, we used the default parameters specified by OpenAI.

---

[1]https://github.com/boxed/mutmut

*3.2.3 Prompts.* Several alternative formulations were explored and evaluated through human-expert testing. While more effective prompting may exist, a systematic comparison of different formulations is out of the scope of this study. We defined individual prompts for input, output, and code prediction tasks. Each prompt follows a consistent general structure and is dynamically populated at runtime with parameters from the dataset, as shown in Figure 3.

Since the function code may require information about helper functions and constants, we include an extra content code section in the prompt. For cross-model compatibility, we used the same prompt and chat template across all models.

*3.2.4 Evaluation of LLM Predictions.* The most meaningful way to assess the quality of LLM predictions is to evaluate their correctness with respect to the specific prediction type. Depending on whether the task involves predicting inputs, outputs, or code, different evaluation strategies are applied:

**Input Prediction:** The predicted input is passed to the known function, which is then executed, and the resulting output is compared with the expected result. If the outputs match, the input prediction is considered valid.

**Output Prediction:** The predicted output is directly compared to the ground truth. An exact match indicates correctness.

**Code Prediction:** The predicted code is executed using the known input, and its output is compared to the expected result. A matching output confirms that the predicted code performs the intended logic.

Note that minor variations in the predicted code can lead to very different execution outcomes. Hence, correctness must be confirmed from runtime behavior, e.g. through tests.

## 4 Evaluation

Our evaluation is designed to measure *functional correctness* rather than surface similarity alone. Depending on the subtask, we therefore assess predictions using execution-based checks (e.g., executing predicted code or validating a predicted input by re-running the reference function) alongside text overlap metrics such as BLEU.

*Evaluation Setup.* Our experiments were conducted on an Nvidia DGX A100 server equipped with two 64-core AMD Rome 7742 processors, 2 TB of RAM, and eight Nvidia A100 GPUs, each with 80GB of VRAM. To run experiments, we packaged the experiment setup as a Docker container using the Nvidia Optimized PyTorch Container 25.01 image [16], which is has access to 16 cores and 1 GPU on the server.

*Evaluation Metrics.* Since multiple valid solutions may exist, text-based metrics alone are insufficient for measuring model prediction quality. In particular, different inputs can legitimately yield the same output, and a function can be implemented in many correct ways. We therefore complement text metrics with *functional tests* tailored to each subtask, capturing surface-level similarity and actual task performance. We evaluated the model's performance using the following metrics:

**Prediction accuracy** measures the ratio of evaluation samples correctly predicted by the model. It provides a clear indication of whether a prediction is valid and aligns with the ground truth, yielding a score from 0 (no samples are valid) to 1 (all predicted samples are correct).

**BLEU score** [22] captures graded overlap and semantic closeness of reference and predicted token sequences, adjusted by a brevity penalty to penalize overly short outputs, yielding a score from 0 (no similarity) to 1 (perfect match). It enables the evaluation of outputs that deviate from the reference but still align in meaning.

## 4.1 RQ1: Can LLMs effectively predict inputs, outputs, or corresponding code?

Given any two components of the LLM-Triad (input, output, and code), can an LLM accurately predict the missing third component? To investigate this question, we evaluated both open-weight models and closed-weight models and compared their prediction accuracy across multiple datasets. The results are presented in Figure 4.

Across all settings, performance varies substantially by dataset, prediction task, and model family, highlighting that triad completion is sensitive to both task formulation and the data's distributional properties.

*4.1.1 Predicting Inputs.* On the Stdlib string dataset, the closed-weight model o4-mini achieves perfect accuracy (100%), correctly reconstructing all test inputs. GPT-5.1 and GPT-4.1 also perform strongly on this dataset, indicating that when transformations are deterministic and string-oriented, LLMs can reliably infer the generating input from the observed output and code. On the other hand, performance decreased on the Voca dataset, with much lower input-prediction accuracy. When looking more closely at test cases and the o4-mini model's responses, it becomes clear that the input is often incorrectly parsed. Since a large part of the test cases involves the `striptag` function, which takes a string of HTML tags and resolves them, the model mistakenly returns tags instead of a string; therefore, many inputs cause an execution error when automatically evaluated. Applying a modification to our automated evaluation, such as converting all predictions to strings, reduces prediction accuracy by 10% overall. We cannot apply such a modification, as it causes incorrect result comparisons and triggers false negative alarms ("" ≠ """"). The results of the input prediction suggest that **LLMs can solve the Input prediction task, even if complex, really well, and can even achieve 100% accuracy**.

*4.1.2 Predicting Outputs.* Output prediction yields slightly lower peak performance than input prediction but exhibits more consistent behavior across datasets. o4-mini achieves the strongest and most consistent performance across datasets (above 69% throughout and often close to 80%), while GPT-5.1 attains the top result on Stdlib (string). Interestingly, fine-tuning improves open-weight models (Phi-4, Qwen3) moderately on some datasets, but the gap to closed-weight models remains substantial. In summary, output-prediction evaluation indicates that **LLMs can simulate program execution to a considerable extent**.

*4.1.3 Predicting Code.* The most challenging task is to predict code from an input and output. While most models fail to achieve more than 30 % accuracy, the best score is achieved on the Python Dataset, showing that the models are more likely to prefer Python. Although this dataset is more challenging than the others because it

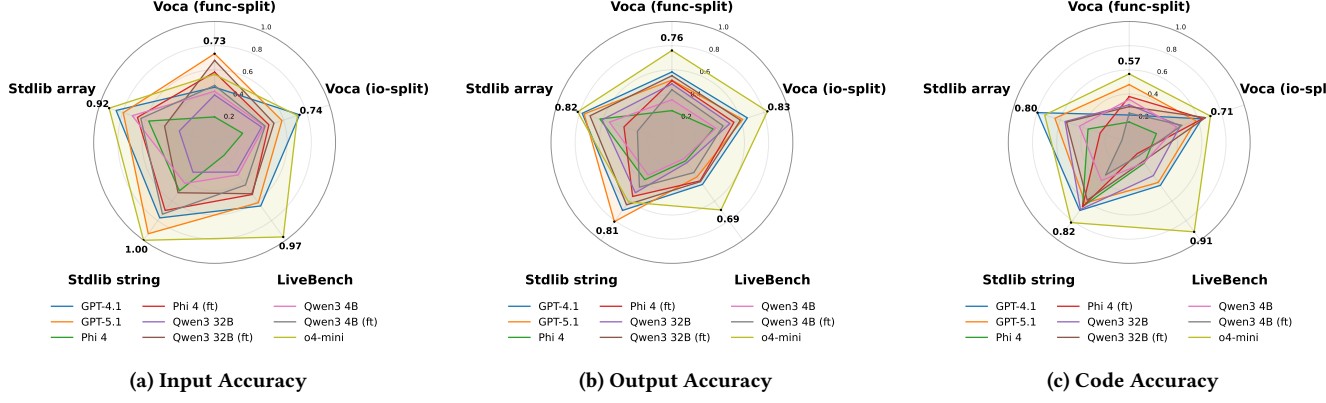

**Figure 4: Prediction accuracy per model across the test sets of each dataset, grouped by prediction task (input, output, and code). FT denotes models fine-tuned on the training data.**

introduces new samples by design, it can still be handled by o4-mini. Overall, the **smaller open-weight models don't perform well enough, even after fine-tuning**.

*Summary.* Our evaluation demonstrates that, in many cases, **LLMs can successfully solve the LLM Triad problem**. In favorable settings, models can achieve near-perfect accuracy, indicating strong capabilities for reasoning about permutations of input, output, and code. However, **the model's performance is highly dataset-dependent and subject-specific**.

## 4.2 RQ2: To what extent does fine-tuning improve the performance of LLM on the LLM-Triad task compared to their pretrained versions?

Pretrained models already demonstrate strong capabilities in code-related tasks. This raises the question: Is additional fine-tuning necessary to achieve optimal performance, or can satisfactory results be obtained without it, thus saving computational resources?

We first analyze the changes in prediction performance between the fine-tuned and base models, focusing on their respective accuracies across tasks (see Figure 4).

Overall, fine-tuning yields the largest average improvement on the input prediction task across all evaluated models. On average, fine-tuned models improve by 17% on input prediction and 9% on output prediction. In contrast, performance on the code prediction task decreases by an average of 2%.

This average decline is primarily driven by substantial performance drops in specific configurations. For example, Qwen3 4B exhibits a −37% decrease on the Stdlib Array dataset. We identify two plausible explanations for this behavior:

- First, *model capacity* appears to play a critical role. Smaller models may struggle to process our comparatively large prompts and the dense information they contain. This hypothesis is supported by the observation that the larger variant of the same model maintains stable prediction rates under identical conditions.

- Second, the *structure* of our fine-tuning data, particularly the use of systematic code mutations, may inadvertently introduce undesirable learning effects. Specifically, we apply mutations to conditional statements (e.g., by modifying if conditions, so they always evaluate to true).

Because similar mutation patterns and token sequences recur frequently in the prompts, smaller models may overfit to these patterns. Instead of learning generalizable strategies, they appear to memorize mutated code fragments and reproduce them in structurally similar tasks. This behavior is particularly problematic when mutated conditions enforce deterministic error paths (e.g., always throwing an exception). During inference, the model may replicate these learned patterns, leading to code that consistently triggers errors and therefore fails to produce the required output. In most cases, this leads to incorrect execution. Despite these limitations, Fine-tuning shows the greatest improvements in the input prediction task. This suggests that input prediction may be underrepresented in standard pretraining corpora. Many LLMs are extensively pretrained on code-reasoning, generation, and execution tasks, whereas input prediction is a comparatively less common objective. Consequently, fine-tuning appears to provide substantial task-specific benefits in this setting.

Finally, we observe a clear correlation between model size and the effectiveness of fine-tuning. **Smaller models exhibit significantly weaker learning dynamics and are more prone to performance degradation**. In contrast, **larger models benefit more consistently from fine-tuning**.

**Phi-4 benefits the most from fine-tuning**, yielding the highest performance increase. Its fine-tuned version even surpasses the base version of Qwen3 32B in performance, despite having approximately half the parameter count. This suggests that architectural factors and pretraining quality may play a more decisive role than parameter scale alone.

Next, we examine text-based evaluation metrics, which provide an additional indicator of how effectively fine-tuning influences model behavior. Although multiple text-based metrics are computed, we focus our analysis on BLEU, as it provides a standardized

measure of n-gram overlap between generated and reference outputs. Figure 5 compares BLEU scores before and after fine-tuning. The overall trend mirrors the results observed in the prediction-based evaluation. In general, we observe modest but consistent improvements across tasks. The largest gains occur in the code prediction and output prediction tasks.

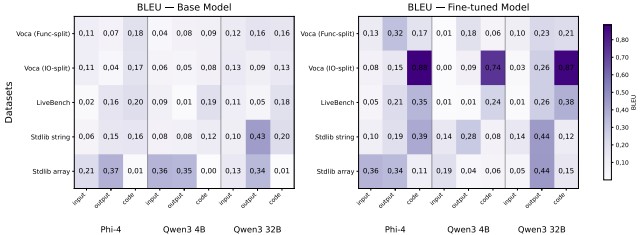

**Figure 5: Heatmap of BLEU values of the base model and fine-tuned model**

The substantial improvement in output prediction is likely due to the task's structure. Because many mutations deterministically introduce specific errors or altered control-flow behavior, the resulting outputs become more predictable. Consequently, lexical overlap between the generated and reference outputs increases, directly benefiting BLEU. In contrast, improvements in input prediction are limited. A plausible explanation is the inherent variability of valid inputs: multiple distinct inputs may satisfy the same functional requirements. As a result, lexical overlap with the reference input may remain low even when the generated input is semantically correct. This observation highlights a limitation of text-based metrics such as BLEU for evaluating input prediction, as these metrics reward surface similarity rather than functional correctness.

We also observe very high BLEU scores on the Voca IO-split dataset. This is expected, since all code samples in this split were already present during fine-tuning. In this setting, the model effectively memorizes and compresses previously seen patterns, resulting in near-identical generations. However, as discussed earlier, high lexical similarity does not necessarily translate into improved prediction accuracy. For the remaining code-prediction tasks, BLEU improvements indicate that the model adapts to the structure and conventions of the provided codebase during fine-tuning. This suggests that **fine-tuning enhances stylistic and structural alignment with the training distribution, even when improvements in execution-based accuracy are more moderate**. Text-based metrics such as BLEU highlight task-specific effects; however, prediction accuracy provides the most reliable measure, as it captures correctness beyond surface-level similarity.

*Summary.* **Fine-tuning improves LLM performance on the LLM-Triad, with the most significant gains observed in input recovery and code generation, particularly in larger models with sufficient capacity**. In contrast, **output prediction exhibits only minor or inconsistent improvements**.

**Table 1: Stdlib String prediction accuracy across fine-tuning strategies (none, Voca, Voca+Stdlib, Stdlib), evaluating cross-codebase knowledge transfer.**

| Model | Predicting Input | | | | Predicting Output | | | | Predicting Code | | | |
|---|---|---|---|---|---|---|---|---|---|---|---|---|
| | Base | $M_{Voca}$ | $M_{Voca,Stdlib}$ | $M_{Stdlib}$ | Base | $M_{Voca}$ | $M_{Voca,Stdlib}$ | $M_{Stdlib}$ | Base | $M_{Voca}$ | $M_{Voca,Stdlib}$ | $M_{Stdlib}$ |
| Qwen3 4B | .419 | .542 | .647 | .733 | .333 | .276 | .495 | .457 | .390 | .380 | .285 | .333 |
| Phi | .495 | .704 | .704 | .695 | .380 | .323 | .552 | .552 | .657 | .276 | .619 | .657 |
| Qwen3 32B | .304 | .733 | .761 | .514 | .514 | .390 | .761 | .638 | .685 | .400 | .657 | .590 |

## 4.3 RQ3: Can the knowledge of a fine-tuned model be transferred/applied to another codebase?

Reusability across multiple codebases is a critical requirement for practical deployment. If a fine-tuned model remains tightly coupled to the dataset it was trained on, its applicability in real-world scenarios becomes limited. Therefore, we investigate whether knowledge acquired during fine-tuning on one codebase can transfer to a structurally related but independently implemented codebase. To evaluate this capability, we designed the Stdlib String dataset to align with the Voca Function split dataset. The Stdlib String dataset is a subset of the Voca Function split by functionality: all functions in the Stdlib dataset are also present in Voca and part of the same train or test set. However, the function implementations differ entirely, providing an ideal setting to assess knowledge transfer beyond memorization of code patterns.

We first evaluate models fine-tuned on Voca directly on the Stdlib dataset to determine whether functional knowledge generalizes across implementations. Subsequently, we perform an additional fine-tuning step on the Stdlib dataset to assess whether prior knowledge facilitates adaptation or whether the model remains biased toward the original training distribution. Finally, e compare this sequential fine-tuning setup with models trained exclusively on Stdlib to assess whether prior exposure to Voca provides benefits.

Table 1 summarizes the results across four configurations: (i) no fine-tuning, (ii) fine-tuning on Voca, (iii) sequential fine-tuning on Voca followed by Stdlib, and (iv) fine-tuning on Stdlib only. Importantly, in all cases, performance is evaluated exclusively on the Stdlib String dataset.

This controlled setup enables a systematic analysis of cross-codebase transferability by isolating the effect of prior training on Voca while keeping the evaluation benchmark constant. Consequently, we can assess whether knowledge acquired from Voca generalizes to alternative implementations of the same functionality or whether the models remain bound to the distribution of their original training data.

The data show mixed results across models and tasks; however, several consistent trends emerge. In general, larger models appear to benefit more from sequential fine-tuning across multiple datasets. This behavior is expected, as higher-capacity models are better able to integrate additional knowledge without fully overwriting previously learned representations.

For the input prediction task, sequential fine-tuning on Voca followed by Stdlib ($M_{Voca,Stdlib}$) yields the strongest performance for Qwen3 32B and Phi. In both cases, performance even exceeds that of task-specific fine-tuning on Stdlib alone ($M_{Stdlib}$). This suggests that

functional knowledge learned from Voca transfers well and can be refined through additional exposure to the Stdlib implementation.

However, this positive trend does not hold consistently across all tasks. For output prediction and code generation, improvements are less stable. In several cases, fine-tuning on Voca alone ($M_{Voca}$) degrades performance compared to the base model on Stdlib. This indicates that knowledge learned from one implementation does not automatically generalize to alternative implementations at the output or code level.

Interestingly, fine-tuning exclusively on Stdlib ($M_{Stdlib}$) often produces competitive or superior results compared to sequential fine-tuning. This suggests that adaptation to the target distribution plays a dominant role, particularly for code prediction, where implementation-specific patterns are crucial. Sequential fine-tuning does not consistently outperform task-specific training, suggesting partial interference between the two training phases.

*Summary.* **Cross-codebase transfer is feasible but highly sensitive to both model capacity and fine-tuning strategy. There is a narrow trade-off between beneficial adaptation and overfitting, and this balance varies across prediction tasks.**

Consequently, optimizing transferability requires careful calibration of the fine-tuning process, as strategies that improve performance for one task may degrade generalization in another.

## 4.4 RQ4: Can open-weight models match the performance of closed-weight models?

While proprietary (closed-weight) models often deliver state-of-the-art results, open-weight models offer greater accessibility and customisation. The question is whether open models can achieve comparable performance on practical tasks.

Based on the previous results, we observe that **open-weight models, even after task-specific fine-tuning, generally do not reach the overall performance levels of closed-weight models**. However, this comparison must be interpreted with caution. Our experimental setup primarily evaluates relatively small open-weight models, which limits the generality of this conclusion.

Notably, even the smallest evaluated closed-weight model, o4-mini, which is assumed to be comparable in size to Phi-4, consistently achieves strong performance and often outperforms larger open-weight models in our benchmark. **Still, open-weight models such as Qwen3 can outperform individual closed-weight models on specific tasks and datasets**, indicating that performance differences are task-dependent rather than absolute. This suggests that factors beyond parameter count—such as pretraining data quality, optimization procedures, and proprietary training strategies—may play a decisive role in determining downstream performance.

*Summary.* **Closed-weight models show superior overall performance, while open-weight models can still compete on specific tasks.**

## 5 Threats to Validity and Limitations

Our study acknowledges several potential threats to the validity of our findings and limitations of our study:

*Internal Validity.* Single code-output pairs per prompt may limit generalization to multi-example contexts and risk overfitting to specific inputs rather than learning generalizable semantics. Uneven sample distribution across functions introduces confounding factors and increases performance variance.

*External Validity.* Our evaluation focuses on string processing, array operations, and coding challenges in JavaScript and Python, limiting generalizability to other languages and domains. As a temporal snapshot, findings may become outdated due to rapid LLM evolution. Specialized domains (e.g., machine learning, systems programming) are not represented.

*Construct Validity.* Single-mutation transformations may not reflect real-world program complexity. Imbalanced test examples per function affect evaluation robustness. Our prompt engineering strategy, while validated by experts, may not generalize across all model architectures. Execution-based correctness, while practical, does not capture code quality dimensions such as efficiency and readability.

*Conclusion Validity.* The limited number of evaluation samples of open-weight models may bias conclusions toward tested architectures. Evaluation metrics capture functional correctness but not code quality; fine-tuning benefits may depend on approach variations. Results reflect a single LLM evolution point; improvements in newer model versions may change findings.

## 6 Discussion and Future Work

Triad completion is feasible, but highly sensitive to dataset properties, prompt context, and the prediction target:

**Predicting inputs** requires *inverse* reasoning and is often underconstrained (many inputs can satisfy the same output).
**Predicting outputs** most closely resembles mental execution, and can be reliable on deterministic tasks.
**Predicting code** is hardest: synthesis generates many correct programs and is particularly sensitive to codebase conventions and prompt format.

Across all settings, closed-weight models consistently perform best, while fine-tuning improves open-weight models mainly on input prediction but can harm code prediction when models overfit to recurring mutation patterns. Cross-codebase transfer is possible, yet fragile: gains are strongest for input prediction and depend on both model capacity and the fine-tuning strategy.

*Future work.* We plan to (i) reduce ambiguity by moving beyond single-example prompts (multiple I/O examples, property-based tests, metamorphic relations), (ii) extend the benchmark toward more realistic program contexts (multi-function modules, state, richer data types, more languages), (iii) integrate hybrid approaches (solver- or execution-guided search/repair for inputs and code), and (iv) evaluate additional quality dimensions beyond functional correctness (robustness, readability, efficiency, and security).

## 7 Data Availability

The data are available at https://anonymous.4open.science/r/Can_LLMs_really_reason_about_Code/README.md

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

Received 20 February 2007; revised 12 March 2009; accepted 5 June 2009

