# OpenReview forum: "Can LLMs really reason about Code? Studying how well LLMs understand the relation between Input, Code, and Output"
_ACM.org/AIWare/2026/Conference — AIware 2026_

### Official Review · Reviewer_YhyS · 2026-03-06

**Rating:** 3
**Confidence:** 3

**Review:**

Strengths

• A comprehensive and systematic study on LLM code semantic reasoning capabilities, decomposing the complex program understanding task into a mutual prediction framework of the "input, output, and code" triad, and evaluating several mainstream open-source and closed-source large code models.

• A set of manually constructed, specialized datasets enhanced by code mutation, designed to ensure low contamination.

• This paper is well written, with clear structure.

Weaknesses

• Inadequate discussion on the generalization boundaries and threats to validity, particularly regarding the limitation to JavaScript/Python and simple, stateless, single-function scenarios compared to complex real-world software.

• Insufficient discussion and experimentation regarding ambiguity in "Input and Output prediction" (e.g., one-to-many mappings); the experiments rely on single-example prompting without comparing few-shot strategies to mitigate ambiguity.

• The open-weight models selected for comparison are relatively small (max Qwen3 32B), lacking comparison with larger-scale open models (e.g., Llama 70B+), which may slightly limit the universality of the conclusions regarding the gap between open and closed models.

Comments for authors

Novelty

• This study systematically proposes the LLM-Triad completion paradigm to evaluate the code reasoning capabilities of LLMs, decomposing program semantic understanding into three types of bidirectional reasoning tasks.  Addtionally, it introduces code mutation techniques to construct low-contamination evaluation datasets, effectively mitigating the pre-training corpus contamination issues prevalent in existing benchmarks.

Significance

• Code reasoning capability is a cutting-edge and highly challenging core topic in the field of software engineering. The findings of this research offer practical guidance for software reverse engineering, automated test case generation, security vulnerability detection, and legacy code refactoring.

• The datasets, evaluation framework, and open-source resources constructed in this paper will contribute to future research on the code reasoning capabilities of LLMs.

Soundness

• This paper examines the LLM-Triad by building four specialized datasets covering multi-language and multi-task scenarios. The experimental design is rigorous. Parallel evaluations with strict variable control are conducted on mainstream open-source and closed-source models using dual-dimensional evaluation metrics. This comprehensive consideration ensures that the findings are robust and applicable.

Nevertheless, this paper could be improved from the following aspects:

• The evaluation scenarios in this paper cover only two programming languages (JavaScript and Python) and focus on simple program scenarios featuring single-function, stateless, and single input-output pairs. However, it is important to point out that in real-world industrial environments, there are numerous system-level programming languages (such as C/C++ and Rust) and complex programs that are multi-module, stateful, and multi-branch. Although the authors briefly mention this limitation in the validity threats section, it is crucial to supplement discussions on:

① the potential limitations of the current evaluation scenarios on the generalization of the research conclusions;

② the potential threats to validity posed by the unincluded programming languages and complex program scenarios;

③ the feasibility and core challenges of extending the current evaluation paradigm to realistic, complex scenarios.

• In the related work section, the paper compares program synthesis studies based on input-output pairs, such as CodeI/O and DeepCoder. However, it does not fully articulate the core differences and complementarities between the proposed LLM-Triad paradigm and mainstream code generation benchmarks (e.g., HumanEval, MBPP) or programming challenge benchmarks (e.g., APPS, CodeContests). I recommend supplementing related discussions to clarify the uniqueness and irreplaceability of the proposed evaluation paradigm, and briefly explaining the limitations of existing mainstream benchmarks to further highlight the innovative value of this research.

• This paper discusses cross-codebase knowledge migration capabilities in RQ3, but experiments are only conducted between two string processing codebases (Voca and Stdlib). I recommend discussing the following aspects:

① the potential boundaries for extending the conclusions of this cross-codebase migration experiment to codebases of different functional types and programming languages;

② the core reasons and underlying mechanisms for the significant differences in cross-codebase migration effects across different tasks, thereby further enhancing the analytical depth of the conclusions.

Replicability

• 	The paper includes a well-organized replication package and a public repository that contains the generated data, which can effectively support other researchers in replicating the experiments of the study.

Presentation

• Overall, this paper is well-structured, with clear exposition of methods, findings, and implications.

• Figure 4 presents the prediction accuracy of various models on different datasets, but lacks clear legend descriptions corresponding to the color schemes of different datasets. Please include a complete legend and add a brief description of the core comparison dimensions in the figure caption to improve readability.

• Table 1 displays the cross-codebase migration results under different fine-tuning strategies. It is recommended to annotate the optimal performance values for each task in the table and supplement a brief summary of the core conclusions in the table notes to help readers quickly capture key findings.

• Please fix the following minor spelling, grammar, and notation issues:
Section 3.1.3: “namly” -> “namely”
Section 4.3: “Finally, e comparee” -> “Finally, we compare”

**Summary:**

This paper examines the capabilities of large language models (LLMs) in code semantic reasoning, focusing on the degree of understanding LLMs have regarding the underlying correlation among code, input, and output. The authors propose the "LLM-Triad" completion problem, which decomposes program semantic understanding into studying the relationships between Input, Code, and Output, requiring the model to predict the third component given any two. To circumvent the benchmark contamination issues arising from pre-training corpora, the authors construct four specialized datasets incorporating code mutation, covering three scenarios across JavaScript and Python: string processing, array manipulation, and coding challenges. Furthermore, comprehensive comparative experiments are conducted on widely-studied open-source models (Phi-4, Qwen3 series) and closed-source models (GPT-4.1, GPT-5.1, o4-mini) to systematically analyze how varying factors—such as model capacity, fine-tuning strategies, and cross-codebase migration—affect task performance. The paper contributes by releasing the code, datasets, and experimental results publicly, providing a standardized benchmark and reusable resources for further research in LLM code reasoning capabilities.

---

> ### Author Response · Authors · 2026-03-21
>
> We thank the reviewer for the careful and constructive feedback. We are encouraged that the reviews agree that the problem is timely, the empirical results are interesting, and the datasets/resources can be useful to the community.
>
> ## 1 Ambiguity in input prediction and the minimal code-prediction setting
>
> Our reviewers are right that *input prediction can be one-to-many*. This is precisely why the main input metric is *functional validity*, not exact lexical match: any predicted input that reproduces the expected output is counted as correct. Thus, ambiguity is part of the task difficulty, but the scoring is designed to be fair to alternative valid inputs.
>
> For code prediction, we agree that a single input-output example is an intentionally minimal and underconstrained specification. The paper should present this more explicitly. The intended task is not full program synthesis, but minimal behavior-conditioned code completion under the same information budget as the other two triad directions. Because this setting is comparatively permissive, the finding that code prediction is still the hardest task is conservative rather than overstated.
>
> In our revision, we will therefore:
>
> - make the underconstrained nature of this task explicit;
> - avoid over-interpreting it as a complete synthesis benchmark;
> - position multiple I/O examples, properties, or metamorphic constraints as natural future extensions.
>
>
> ## 2 Model size
>
> We agree that the conclusion should be scoped to the practically fine-tunable open-model regime studied here (4B–32B, single-GPU-friendly setup), not the entire open-weight ecosystem. This choice was deliberate because the paper also studies adaptation. A revision can simply make this scope explicit.
>
> Even with that narrower statement, the result remains useful: In the regime that many researchers and practitioners can realistically fine-tune, closed models are still the strongest overall baselines, while open models remain competitive on selected tasks.
>
> Future work can explore larger models, using this work as a baseline to enable more efficient experimentation and avoid unnecessary resource usage.
>
> ## 3 Discussion Adjustments
>
> We acknowledge that the current discussion is relatively sparse.
> We will strengthen it to improve clarity, analytical depth, and robustness of the conclusions.
>
> First, we will address **limitations of the evaluation scenarios**. The use of small and constrained code samples may limit generalizability. Future work will include larger datasets and more expressive models to validate and extend the findings.
>
> Second, we will discuss **threats to validity from limited programming language coverage and simple program structures**. The current evaluation excludes multiple languages and complex real-world scenarios. Future work will expand to more languages and diverse, complex codebases to improve generalizability.
>
> Third, we will consider **feasibility of extending the evaluation to realistic settings**. While the current framework works in controlled environments, real-world scaling introduces challenges such as code complexity, dependency management, and environment variability. Future work will explore full libraries and end-to-end systems.
>
> Finally, we will clarify the **boundaries of cross-evaluation migration**. We will define when results transfer across settings and where such transferability breaks down, to better understand the limits of generalization.
>
> ## 4 Novelty relative to prior work
>
> The contribution is instead the unified triadic framing and matched evaluation setup:
>
> - one benchmark covering all three directions jointly: input from code+output, output from code+input, and code from input+output;
> - one execution-based evaluation protocol across these directions;
> - mutation-augmented datasets to reduce contamination and introduce controlled semantic variation;
> - an analysis of task-specific fine-tuning and cross-codebase transfer within the same framework.
>
> In our revision, the related work section will clearly position this contribution, regarding the Dataset generation.
> The intended differentiation is that programming-by-example work mainly studies code synthesis from I/O, runtime-behavior work focuses on execution/output reasoning, and CruxEval-style benchmarks emphasize input/output reasoning rather than the full triad of code reconstruction, fine-tuning, and transfer in a single common setup.
>
> ## 5 Presentation changes
>
> The requested improvements are straightforward and can be incorporated in a revision:
>
> - Fix the noted typos and minor grammar issues.
> - Improve Figure 4 readability with a clearer legend/caption, and possibly a simpler alternative visualization;
> - Annotate best values in Table 1 and summarize the transfer takeaway in the caption/note.
>
> Thank you!

---

### Official Review · Reviewer_e762 · 2026-03-07

**Rating:** 2
**Confidence:** 4

**Review:**

**Pros**
- Benchmarking the code reasoning capabilities of LLMs is a timely and relevant topic.
- The paper is overall well written.
- The evaluation includes several open- and closed-weight LLMs, as well as multiple datasets covering two programming languages.

**Cons**
- Limited consideration of the existing literature.
- The novelty with respect to prior work is not adequately articulated.
- Some methodological aspects remain unclear.

**Evaluation**

1. Benchmarking the code reasoning capabilities of LLMs is currently a very active research area. Unfortunately, the authors do not adequately cover the existing literature, nor do they clearly explain the key differences between their work and prior studies. In particular, the three problems involved in the LLM-Triad have already been considered in earlier papers, including:
	- Li, Wen-Ding, and Kevin Ellis. “Is Programming by Example Solved by LLMs?” Advances in Neural Information Processing Systems 37 (2024).
	- J. Chen, Z. Pan, X. Hu, Z. Li, G. Li, and X. Xia. “Reasoning Runtime Behavior of a Program with LLM: How Far Are We?” 2025 IEEE/ACM 47th International Conference on Software Engineering (ICSE).
	- Gu, Alex, et al. “CruxEval: A Benchmark for Code Reasoning, Understanding and Execution.” arXiv preprint arXiv:2401.03065 (2024).
	- Ruiyang Xu, Jialun Cao, Yaojie Lu, Ming Wen, Hongyu Lin, Xianpei Han, Ben He, Shing-Chi Cheung, and Le Sun. “CRUXEVAL-X: A Benchmark for Multilingual Code Reasoning, Understanding and Execution.” Proceedings of the 63rd Annual Meeting of the Association for Computational Linguistics (2025).

2. This limited acknowledgment of the existing literature makes it extremely difficult to assess the originality of the work, as the authors do not sufficiently articulate what is novel in their study compared with the most relevant prior work.

3. The results are somewhat interesting, however their significance appears limited in terms of advancing knowledge beyond what is already known from prior work.

4. The overall quality of presentation is good, but there are some aspect issues that could be improved:
	- The spider plots used in the figure 4 make the results somewhat difficult to read.
	- I found the explanation of the issue related to the striptag function in Section 4.1.1 somewhat hard to follow.
	- Page 4, line 365: “namly” should be corrected to “namely”.
	- Page 7, line 787: “e” should be corrected to “we”.

5. Some methodological aspects also remain unclear:
	- It is not clear how the authors ensure the quality of the resulting program-input-output triples after applying mutation, especially with respect to code comments.
	- For LiveCodeBench, the authors sample 80 distinct functions and then generate ten mutated variants for each. It is unclear why they did not simply sample a larger number of programs in the first place.
	- According to the discussion of threats to validity, the paper appears to consider only a single input and a single output per program for each prediction task. If this is indeed the case, the code prediction task seems rather weak as a benchmark for evaluating LLM reasoning abilities.

**Summary:**

The paper aims to benchmark the code reasoning capabilities of a selection of open-weight and closed-weight models. Specifically, the authors focus on what they call the LLM-Triad, namely the ability of a model to predict: (i) the input from the code and output, (ii) the output from the code and input, and (iii) the code solely from the input and output.

---

> ### Author Response · Authors · 2026-03-21
>
> We thank the reviewer for the careful reading and the concrete suggestions.
> The positive assessment of the topic, writing quality, and breadth of evaluation is appreciated.
> The main concerns raised—positioning with respect to prior work and several methodological clarifications—are well taken and can be addressed through a focused revision without altering the paper’s core contribution.
>
> ## 1 Novelty relative to prior work
> The paper does not claim that each of the three prediction problems is individually new.
> The contribution is instead the _unified triadic framing_ and matched evaluation setup:
>
> - one benchmark covering all three directions jointly:
>   * input from code+output,
>   * output from code+input, and
>   * code from input+output;
> - one execution-based evaluation protocol across these directions;
> - mutation-augmented datasets to reduce contamination and introduce controlled semantic variation;
> - an analysis of task-specific fine-tuning and cross-codebase transfer within the same framework.
>
> We appreciate the reviewer's concern regarding the related work; we will explicitly discuss the cited works and clarify the distinction.
>
> The intended differentiation is that programming-by-example work mainly studies code synthesis from I/O, runtime-behavior work focuses on execution/output reasoning, and CruxEval-style benchmarks emphasize input/output reasoning rather than the full triad of code reconstruction, fine-tuning, and transfer in a single common setup.
>
> ## 2 Why the results add value
>
> The main empirical value is that a _single controlled protocol reveals asymmetries that are difficult to see when tasks are studied separately_.
>
> In particular, the results show that:
>
> - output prediction is the most stable task.
> - code prediction is the hardest.
> - input prediction benefits most from fine-tuning.
> - cross-codebase transfer is feasible but strongly task-dependent.
>
> Even if the subproblems have appeared individually before, these comparative findings arise from placing them into a single directly comparable benchmark.
> The paper thus also adds value as an empirical study of triadic completion in controlled settings.
>
> ## 3 Quality of mutated program–input–output triples
>
> Let us clarify this aspect.
> The triples are grounded in actual execution of the mutated program: after mutation, the mutated function is run on the selected input, and the observed runtime result becomes the output label.
> Regarding comments, these are only auxiliary prompt context for helper code/imports; they are not used to define labels.
> Mutations are applied to the target function under study, not to helper functions.
> Thus, comments do not affect triple validity; at worst, stale comments would add prompt noise.
> We will clarify this in the paper.
>
> ## 4 Why we used only 80 LiveCodeBench functions plus mutations
>
> Our choice balances functional diversity and controlled variation.
> In the selected LiveCodeBench split, multiple samples correspond to alternative implementations of the same task.
> One representative per functionality was selected to avoid over-weighting tasks with many implementations.
> Mutations then provide systematic local variations around the same underlying behavior.
> We will clarify this rationale.
>
> ## 5 Single input/output pair and the code-prediction task
>
> The reviewer is correct that each prompt uses a single concrete I/O example.
> This makes code prediction a sparse and intentionally difficult benchmark.
> The intended claim is not that this is a full program-synthesis benchmark, but that it serves as a minimal behavioral-completion test that keeps the three triad directions comparable.
>
> Two points are important:
>
> - Code predictions on semantical equivalence and functionally, not by syntactical similarity, so implementation variations are allowed.
> - We will explicitly state that single-example code prediction is a conservative test of behavioral reasoning, not the final word on synthesis ability.
>
> ## 6 Presentation
>
> Thank you for your presentation suggestions!
>
> - We will replace the radar plots with a more readable visualization if page spacing allows it, or at a minimum, add clearer legends and captions.
> - We will rewrite the striptag explanation around a concrete example.
> - We will correct all noted typographical issues.
>
>
> ## 7 Summary
>
> In summary, the concerns raised mostly call for clearer framing and clearer explanation, rather than changes to the core study.
> The paper’s main value lies in the unified triadic benchmark, execution-based evaluation, mutation-aware dataset design, and the comparative findings on task difficulty, fine-tuning, and transfer.
> Following the reviewer's suggestion, the interpretation will be narrower and more precise, and the contribution will be stronger and better grounded. Thank you!

---

### Official Review · Reviewer_YbrV · 2026-03-11

**Rating:** 3
**Confidence:** 3

**Review:**

Pros:
+ Important and timely research.
+ Framing program understanding as a triadic completion task is interesting and intuitive.
+ The paper builds four datasets that can be leveraged for future research
+ Results are interesting and not trivial


Cons:
- A model can guess the missing part of the triad without deeply understanding the program.
- Input prediction can be naturally ambiguous, because more than one input can produce the same output. This makes the task harder to evaluate fairly
- Some fine-tuned models show large performance drops (e.g. Qwen3 4B drops by 37% )

Detailed comments to the authors:

Novelty

+ The paper is novel in the formulation of its triadic framing of program understanding: instead of testing only whether a model can predict outputs or generate code, it studies all three directions among input, output, and code. This introduces a clearer evaluation lens than many standard code-generation papers, and it is strengthened by the inclusion of four datasets across JavaScript and Python, additionally a transfer setting between Voca and Stdlib. As an empirical study, this is a meaningful and well-motivated contribution.

- At the same time, the novelty is more in the study design and benchmark framing than in a new technical mechanism. The related-work section already discusses prior work in multiple dimensions including input-output reasoning, behavior modeling, execution-aware code modeling, and code reasoning, so the contribution is not that the paper opens an entirely new area, but that it brings these ideas together into one unified empirical question. I think that is still a key contribution, but the paper would benefit from stating this more precisely.

Significance

+ The paper is important because it asks a question that is broader than typical code generation benchmarks, i.e., whether LLMs can track relationships between program structure and behavior. That matters for how we interpret current code LLM capabilities, and the results are useful and interesting. The study finds an interesting pattern where closed-weight models are strongest overall, output prediction is the most stable task, code prediction is the hardest, and transfer across codebases is possible but inconsistent. These are practical findings that future studies can build on, and they make the paper more valuable than a narrow benchmark report.

- However, the significance should be stated within the paper’s actual scope. The authors themselves acknowledge that the study focuses on string processing, array operations, and coding challenges in JavaScript and Python, with single-example prompts and single-mutation transformations. Because of that, I read the paper as strong evidence about triad completion in controlled settings, not yet as a broad answer to whether LLMs “really reason about code” in the full software-engineering sense. The title is ambitious relative to the actual study. A realistic interpretation is that the paper gives useful evidence about how models behave on small, self-contained program understanding tasks. I would encourage the authors to narrow the claim slightly so that the significance comes from the clarity of the empirical findings, not from a broader promise than the study can support.

+ Another positive point is that the transfer analysis adds value because it asks whether knowledge learned on one implementation transfers to another implementation of the same behavior. That is important for judging whether models learn something functionally meaningful. But the study also shows that transfer is uneven and depends on task and model capacity, so the finding here is not that transferability works,  but rather it is partial and can be task-sensitive. That can be a useful empirical message.

Soundness

+ The study is reasonably sound for its intended scope. It evaluates on separate test data, leverages multiple datasets, compares open- and closed-weight models, and does not rely only on typical metrics where the paper explicitly treats functional correctness as the main signal and uses execution-based evaluation where possible. I also appreciated that the authors include a proper limitations section and do not hide the fact that performance is highly dataset-dependent. As a study, this gives the work a fair amount of credibility.

- My main concern is that some conclusions are still easier to support than others. The strongest supported conclusion is that LLMs can often complete the code, input, or output triad in controlled cases, with strong variation by task and dataset. The less supported conclusion is the broader claim about “reasoning.” Within the paper’s scope, there are several acknowledged factors that make the evidence narrower such as prompts containing only one example, input prediction can be underconstrained because many inputs can match the same output, and code prediction is the hardest task and remains weak for most models. So I think the empirical evidence is useful, but the interpretation should stay close to what was actually evaluated.

- There are also a few places where I wanted clarification. First, in the fine-tuning section, the paper says that fine-tuning improves input prediction by 17% and output prediction by 9% on average (line 624), but decreases code prediction by 2% on average. Later, the summary says that the most significant gains are seen in input recovery and code generation (lines 749-752). These two statements do not read as fully consistent, so I would like the authors to clarify whether the code generation gains performance improvements are relevant to certain larger models only, or to some subset of settings rather than the overall average. Also, It would be interesting to see a discussion on whether we need other alternatives to improve performance of code predication tasks given that finetuning drops the performance.
Second, the paper notes a concrete evaluation issue on Voca input prediction, where parsing problems around striptag predictions affect the measured result and even a possible evaluation fix would lower overall accuracy by 10%. That makes me wonder how much of the observed difficulty in that setting is model weakness versus evaluation artifact. A small manual error analysis would make the study more convincing here, even if only on a sample of cases.

- Third, the transfer results are interesting but mixed where input prediction benefits most from sequential fine-tuning, while output prediction and code generation are less stable, and in some cases fine-tuning on Voca alone hurts performance on Stdlib. That does not weaken the study, but it does mean that the transfer story should be framed carefully. The finding on partial transfer with interference is not a general claim that learned functional knowledge transfers robustly across codebases.

Questions:
- What is the concrete takeaway for future software engineering research or practice from these results?
- What should future SE benchmarks prioritize based on this study?
- What should practitioners conclude about fine-tuning from this study?

**Summary:**

This paper studies whetherLLMs genuinely understand program behavior, beyond their strong performance in code generation. The authors frame program understanding as a triadic reasoning problem over code, input, and output, asking whether an LLM can recover one component when given the other two, i.e., predicting the input from code and output, predicting the output from code and input, and predicting the code from input and output. To evaluate this, they build four datasets spanning string processing, array operations, and coding challenges in both JavaScript and Python, and they increase task difficulty through mutation-based program transformations. Across experiments on open-weight models such as Phi-4 and Qwen3, as well as closed-weight GPT-family models, they find a consistent performance gap in favor of closed-weight systems. They also report that output prediction is generally the most stable task, code prediction is the hardest, and cross-codebase transfer is possible but sensitive to model capacity and fine-tuning strategy.

---

> ### Author Response · Authors · 2026-03-21
>
> We thank the reviewer for the careful and constructive feedback.
> We are encouraged that the reviews agree that the problem is timely, the empirical results are interesting, and the datasets/resources can be useful to the community.
> The paper is intended primarily as an **empirical benchmark/study-design contribution** rather than as a new reasoning algorithm.
> We will address the concerns through clearer positioning, sharper discussion of prior work, and modest presentation changes.
>
> ## 1. “Guessing” vs. understanding
>
> We agree that no behavioral benchmark can fully rule out shortcutting.
> The paper does not need to claim otherwise.
> Its goal is to test whether models capture code–input–output relationships strongly enough to complete the missing element under controlled holdout settings.
>
> Several design choices already reduce trivial memorization:
> - mutation-based generation of unseen variants;
> - held-out evaluation;
> - execution-based validation rather than surface-only matching;
> - cross-codebase evaluation on semantically aligned but independently implemented functions.
>
> If the model's performance were driven mainly by superficial guessing, one would expect much flatter results across tasks and datasets.
> Instead, the paper reveals strong, systematic asymmetries.
> We will revise the paper accordingly, elaborating on this point.
>
>
> ## 2. Ambiguity in input prediction
>
> *Input prediction can indeed be one-to-many*.
> That is why the main input metric is *functional validity*, not lexical match: any predicted input that reproduces the expected output is counted as correct.
> Thus, ambiguity is part of the task difficulty, but the scoring is designed to be fair to alternative valid inputs.
>
> Thus, we will:
> - make the underconstrained nature of this task explicit; and
> - avoid over-interpreting it as a complete synthesis benchmark.
>
> ## 3. Fine-tuning: gains, drops, and the current wording inconsistency
>
> The reviewers identified an important wording issue. The paper currently mixes execution-based accuracy and BLEU/structural similarity. The accurate aggregate statement is that fine-tuning helps input prediction the most. It gives modest average gains for output prediction, and it does not improve average functional code-prediction accuracy overall.
>
> The sentence suggesting that the strongest overall gains are in “input recovery and code generation” is therefore too broad and can be revised.
> The current results support the idea that some larger-model settings show code-generation improvements in BLEU/structure or in specific transfer conditions, while the average execution-based code accuracy remains flat or slightly worse.
>
> The large drops in some small models are also informative rather than incidental: they suggest that mutation-heavy fine-tuning can induce negative transfer or overfitting when capacity is limited. This is a useful practical finding, and the revision will it more clearly as such.
>
> We will also add a concise practitioner takeaway: *Fine-tuning is promising for input recovery and codebase adaptation, but behavior-only code prediction is much more sensitive to model size and data design.*
>
> ## 4. Cross-codebase transfer
>
> We will be happy to frame the transfer section more narrowly, claiming that partial transfer occurs within semantically aligned but independently implemented codebases,not robust transfer across arbitrary languages or software systems.
>
> The observed task differences are meaningful:
> - Input prediction transfers best because it depends more on shared functional regularities;
> - Code prediction is most sensitive to implementation style, helper structure, and library-specific conventions;
> - Output prediction lies between these extremes and can still suffer interference after sequential fine-tuning.
>
> We will state these boundaries more explicitly and expand the limitations discussion to explain why extension to multi-module, stateful, or lower-level languages is non-trivial (inter-procedural dependencies, hidden state, larger context windows, and more complex execution behavior).
>
> ## 5. Concrete takeaways for software engineering
>
> We will make the implications of our work explicit:
> - For benchmark design: SE benchmarks should cover all three directions, not only NL→code or code+input→output.
> - For evaluation: execution-based correctness is essential; text similarity alone is not enough for inverse reasoning or code reconstruction.
> - For contamination control: mutation-based or otherwise freshness-aware data construction matters.
> - For ambiguity control: inverse tasks should ideally use multiple I/O examples, properties, or metamorphic relations.
> - For practitioners: fine-tuning should not be assumed to help uniformly.It is useful for input recovery and target-codebase adaptation; for sparse behavior-conditioned code synthesis, especially with smaller open models, it can hurt as much as help.
>
> These takeaways are actionable even without claiming broad industrial generalization.Thank you!

---

### Author Response · Authors · 2026-03-21

We thank the reviewers for their valuable and constructive feedback! We appreciate the reviewers’ time; their suggestions will significantly improve the paper.

We will address each reviewer’s feedback individually.
We will incorporate the following general improvements, following the suggestions:

- Refine the paper title for greater clarity and precision
- Revise and expand the related work section
- Strengthen the discussion section with more depth and clarity
- Improve figures and tables to better highlight results
- Clarify several methodological aspects, including:
  - Mutations and comments
  - LiveCodeBench setup
- Clarify the role of string tags
- Clarify fine-tuning results
- Enhance the discussion with clearer key takeaways and real-world implications